# Heterogeneous associations of a mobile health-based disease management program on uncontrolled hypertension: A target trial emulation study

**Masashi Kanai[1,2], Takahiro Miki[1]*, Takuya Toda[1], Yuta Hagiwara[1], Takaaki Ikeda[1,3,4]**

**1** Insight Lab, PREVENT Inc., Aichi, Japan, **2** Institute of Transdisciplinary Sciences for Innovation, Kanazawa University, Kanazawa, Japan, **3** Department of Health Policy Science, Graduate School of Medical Science, Yamagata University, Yamagata, Japan, **4** Department of International and Community Oral Health, Tohoku University Graduate School of Dentistry, Sendai, Japan

* miki.takahiro@prevent.co.jp

## Abstract

Long-term effectiveness of digital health interventions for hypertension remains unclear, particularly regarding individual variability in treatment response. This study examined the association of a mobile health-based disease management program for uncontrolled hypertension and assessed treatment effect heterogeneity using a target trial emulation framework. We analyzed health checkup data of employees June 2021–December 2023. Individuals with hypertension, diabetes, or dyslipidemia were invited to participate in a six-month mobile health-based disease management program incorporating lifestyle tracking via a mobile application and remote behavioral coaching. We compared the following two treatments using a target trial emulation framework: mobile health-based disease management program combined with conventional treatment, versus conventional treatment alone. The primary outcome was uncontrolled hypertension at the one-year follow-up (systolic ≥140 mmHg or diastolic ≥90 mmHg). We estimated average and individual treatment effects using outcome regression based on the G-formula with ensemble machine learning methods for model specification. Clustering analysis was used to identify heterogeneous subgroups and potential effect modifiers. Mobile health-based disease management program was associated with a 5.2% (95% confidence interval: 4.4% to 6.0%) lower prevalence of uncontrolled hypertension compared with conventional treatment. Treatment response varied, with greater benefits observed in individuals with a strong intention to improve lifestyle habits, higher diastolic blood pressure, and more favorable behavioral and metabolic characteristics. Age was associated with benefit, though it had relatively lower importance. Participation in a mobile health-based disease management program was associated with better blood pressure control over one year. The substantial variation in treatment effectiveness highlights the need for personalized digital health strategies.

which permits unrestricted use, distribution, and reproduction in any medium, provided the original author and source are credited.

**Data availability statement:** The data underlying this study contain potentially identifiable participant information and cannot be made publicly available due to ethical and legal restrictions. Data are available from the Data Administration Office at PREVENT Inc. (info@prevent.co.jp) for researchers who meet the criteria for access to confidential data.

**Funding:** PREVENT Inc. provided support for this study in the form of salaries for authors T.M., T.T., and Y.H., and compensation for non-regular staff M.K. and T.I. The funders had no role in study design, data collection and analysis, decision to publish, or preparation of the manuscript.

**Competing interests:** I have read the journal's policy and the authors of this manuscript have the following competing interests: T.M. and T.T. are employees of PREVENT Inc. Y.H. is the founder and CEO of PREVENT Inc. M.K. and T.I. are non-regular staff members of PREVENT Inc.

## Author summary

This study examined whether a mobile health program can help maintain blood pressure control over one year, and whether benefits differ across individuals. Employees in Japan were invited to use a mobile app to track daily habits—such as weight, activity, diet, and blood pressure—and to receive remote coaching from health professionals; outcomes were compared with those receiving standard care only using methods from causal inference to make the comparison as fair as possible. After one year, about 5% fewer participants in the mobile health program had uncontrolled hypertension compared with those receiving standard care alone. Benefits were not uniform: individuals who already intended to improve their lifestyle, who began with higher diastolic blood pressure, and who showed more favorable day-to-day habits and routine health measures tended to benefit more. By contrast, age was linked to benefit but played a smaller role than these modifiable factors. These findings suggest that mobile health can meaningfully support blood pressure control and that tailoring program intensity and content to each person's readiness and needs may enhance effectiveness.

## Introduction

Hypertension is a major global public health concern and a leading risk factor for cardiovascular disease [1–3]. Effective blood pressure management requires both pharmacological treatment and lifestyle modifications [4,5]. However, traditional hypertension management programs often struggle with patient adherence [6,7]. Advancements in digital health technologies, including mobile health (mHealth) solutions, have expanded the use of smartphone applications, wearable devices, and remote monitoring systems for blood pressure management [8–10].

Systematic reviews and meta-analyses have consistently shown that digital health and mHealth interventions—such as short message services (SMS), smartphone applications, and web-based platforms—significantly reduce systolic and diastolic blood pressure [11–20]. Despite these promising findings, the effectiveness of digital health interventions varies across studies, influenced by individual characteristics. Previous studies have indicated that factors such as age, baseline blood pressure, and familiarity with digital tools affect outcomes [11,15,21]. Moreover, evidence on their long-term effects remains limited, as most studies have follow-up periods of only 3–12 months [12,18–20,22]. These gaps highlight the need for further research to assess treatment heterogeneity and evaluate the sustained benefits of digital health interventions over extended follow-up periods. To address these gaps in knowledge, this study examined the heterogeneity in the association between an mHealth-based disease management program [23,24] and uncontrolled hypertension over a one-year follow-up using the target trial emulation framework.

## Methods

### Setting and participants

In this study, we used "target trial emulation" framework, which emulates a target randomized control trial design [25], combined with a roadmap for causal inference to enhance the quality and interpretation of observational studies [26]. Table 1 represents the core components of the target trial protocols and emulation strategy. This study used data from the Specific Health Checkup and medical claims records [27,28] provided by the companies employing the participants. Japan's public health insurance system is a compulsory, publicly funded program offering affordable medical services to all residents, including both employed and unemployed individuals. It contains two main types: 1) Employer-based health insurance—covers employees and their dependents, provided through employers. 2) National health insurance—covers self-employed individuals, retirees, and those not covered by employer-based insurance, managed by municipalities and prefectures.

Under this system, all insured individuals have access to medical care, including preventive health services. The Specific Health Checkup, a government-mandated annual health screening program, focuses on preventing lifestyle-related diseases, particularly metabolic syndrome. As a part of the public health insurance system, it targets individuals aged 40–74 years enrolled in either Employer-based or National health insurance [27,28].

We analyzed data from participants who met the hypothetical eligibility criteria from June 2021 to December 2023. Fig 1 illustrates the study timeline, from the start of follow-up (time zero) to outcome assessment.

**Table 1. Specification of the target trials enrolling patients.**

| Protocol components | Target trial specification | Emulation of the target trial |
|---|---|---|
| Eligibility criteria | Inclusion: Participants diagnosed with hypertension, diabetes, or dyslipidemia who were either undergoing pharmacological treatment or had a history of coronary artery disease or stroke.<br>Exclusion: Participants with medical conditions that could interfere with participation, including severe cardiac diseases (e.g., arrhythmia, cardiomyopathy), end-stage renal disease, or mental disorders; individuals taking medications such as cardiotonic agents, immunosuppressants, or antipsychotics; and individuals deemed unsuitable by the health insurance association based on predefined criteria. | Same as the target trial |
| Treatment assignment | Individuals were randomly allocated following the eligible assessment. | Individuals were nonrandomly assigned to the treatment arms described below. Randomization was emulated using G-formula, with adjustment for baseline covariates. |
| Treatment arms | Receipt of a mobile health-based disease management program+conventional treatment vs. conventional treatment [Key components of the mobile health-based disease management program]<br>(1) 12 phone sessions over six-months with healthcare professionals, plus chat support<br>(2) Use of a mobile app to track lifestyle data and review educational content<br>(3) Personalized coaching and health goal-setting focused on behavior change | Same as the target trial |
| Follow-up period | 365 days after the treatment assignment. | 275 to 455 days after the treatment assignment. |
| Outcomes | Primary outcome: uncontrolled blood pressure, defined as systolic ≥140 mmHg or diastolic ≥90 mmHg.<br>Secondary outcomes: changes in systolic and diastolic blood pressure from baseline to follow-up. | Same as the target trial |
| Causal contrasts of interest | Intention-to-treat | Same as the target trial |

 

PLOS Digital Health

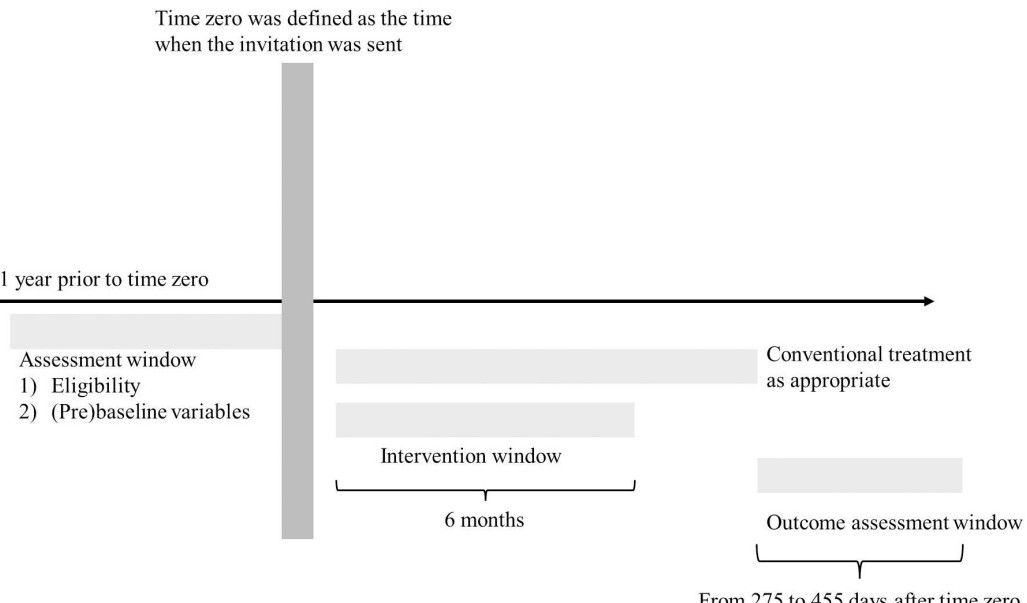

**Fig 1. Schematic overview of time zero, treatment assignment, and follow-up in the target trial emulation.**

The study followed the Strengthening the Reporting of Observational Studies in Epidemiology (STROBE) reporting guideline for cohort studies.

### Inclusion and exclusion criteria

This study included participants diagnosed with hypertension, diabetes, or dyslipidemia who were either receiving medication or had a history of coronary artery disease or stroke. These eligibility criteria reflected the actual enrollment policy of the original mHealth-based disease management program, which targeted people with common lifestyle-related diseases rather than hypertension alone. Because these conditions frequently coexist and share overlapping pathophysiological and behavioral risk factors, the program was implemented as a disease management program for lifestyle-related conditions, with the aim of supporting integrated management and preventing progression of cardiovascular complications, rather than as a single-disease intervention. Eligible individuals were identified through the Specific Health Checkup and employer-provided medical claims data and were invited to participate in the mHealth-based disease management program.

Exclusion criteria included participants with medical conditions that could interfere with participation, such as severe cardiac diseases (e.g., symptomatic arrhythmias requiring close monitoring, dilated and hypertrophic cardiomyopathy, and other unstable cardiac conditions), end-stage kidney disease, or mental disorders; individuals taking medications such as cardiotonic agents, immunosuppressants, or antipsychotics; and those deemed unsuitable by the health insurance association based on predefined criteria. These exclusion criteria were operationalized using predefined contraindication codes for the remote, app-based disease management program and were intended to identify individuals for whom safe and stable participation, including adequate adherence, might be difficult. A complete list of exclusionary diagnoses is available at the GitHub link (https://github.com/PREVENT-Inc/MyscopeMasterList/tree/master/Out_of_scope). Additionally, participants with follow-up periods of less than 275 days were excluded, accounting for the time from invitation to the completion of the six-month intervention.

## Treatment regimen

In Japan, hypertension treatment is provided under the universal health coverage system, where insured individuals visit medical clinics, receive a diagnosis from a physician, and, if necessary, are prescribed antihypertensive medication [29,30]. This standard approach is referred to as "conventional treatment" in this study.

The mHealth-based disease management program, Mystar, was implemented alongside conventional treatment with approval from participants' attending physicians. The program lasted six months and included 12 phone call sessions with healthcare professionals, supplemented by chat messaging between sessions. Participants were provided access to the mHealth app, where they entered lifestyle data such as body weight, blood pressure, physical activity, salt intake, and food photos, enabling healthcare professionals to monitor progress and provide personalized coaching. Within the app, participants could track lifestyle behaviors, access personalized health goals and to-do lists, communicate with healthcare providers, and review educational materials on hypertension and non-communicable disease risk management. The intervention targeted exercise, diet, sleep, alcohol intake, smoking cessation, and stress management through individualized health plans. Further details on program structure, engagement strategies, and adherence are available in a previous study [23,24]. Therefore, the intervention group received both the mHealth-based disease management program and conventional treatment as appropriate, while the control group received only conventional treatment in this study (Fig 1).

## Covariates

The following covariates were assessed at the time of eligibility evaluation: age (continuous); sex (male/female); diagnosis of hypertension, diabetes, or dyslipidemia, or related medication prescription in the past 12 months (yes/no); smoking status (yes/no); drinking status (every day, sometimes, rarely, never); weight change since age 20 (yes/no); intention to improve lifestyle habits (yes/no); engaging in exercise that causes light sweating (≥30 minutes per session, ≥ 2 times/week, for over one year) (yes/no); walking or equivalent physical activity (≥1 hour/day) (yes/no); eating speed (fast/normal/slow); having dinner within 2 hours before bedtime (yes/no); skipping breakfast ≥3 times/week (yes/no); getting sufficient rest through sleep (yes/no); body mass index (continuous); triglycerides (continuous); High-density lipoprotein (HDL)/Low-density lipoprotein (LDL) cholesterol (continuous); diastolic/systolic blood pressure (continuous); Glutamate Oxaloacetate Transaminase (continuous); glutamate pyruvate transaminase (continuous); Gamma-GTP (continuous); hemoglobin A1c (continuous); Charlson Comorbidity Index (continuous); number of days since the last health checkup (continuous). Medical history data were obtained from medical claims records, while lifestyle and clinical data were extracted from the most recent health checkup before eligibility assessment.

## Outcome

The primary outcome was uncontrolled hypertension control, defined as systolic blood pressure ≥140 mmHg or diastolic ≥90 mmHg. Blood pressure was assessed annually during the health checkup. The secondary outcomes were the differences in systolic and diastolic blood pressure between baseline and follow-up. Blood pressure was assessed annually during the Specific Health Checkup. In principle, blood pressure was measured twice using a standard sphygmomanometer or automated device on the right arm after a 5-minute rest in the sitting position, and the average of the two readings was recorded. This procedure follows the standard protocol of Japan's Specific Health Checkup program, which has been widely used in large-scale epidemiological studies published internationally [31,32].

## Statistical analysis

The primary causal contrast of interest was the intention-to-treat analysis. We conducted a descriptive analysis based on the treatment status. Subsequently, we estimated the average treatment effect (ATE) and individual treatment effect (ITE). We employed the following three-step analytic approach: First, we estimated the ITE of the mHealth-based disease

management program on the outcome using the G-formula (outcome model), implemented with data-adaptive model specification via the SuperLearner ensemble machine learning algorithm [33]. Candidate SuperLearner algorithms included generalized linear models, random forests, neural networks, multivariate adaptive regression splines, and extreme gradient boosting models. ITE was calculated as the risk difference between the probability of uncontrolled blood pressure in the treatment and control groups under a counterfactual framework. A negative ITE indicates a lower risk of uncontrolled blood pressure. In addition, the corresponding risk ratio was derived as the ratio of these probabilities. For interpretability, the number needed to treat (NNT) was calculated based on the absolute risk difference. Then, the ATE was estimated using bootstrap resampling to quantify uncertainty. Subsequently, we assessed the heterogeneity of the effect via cluster modeling to identify effect modifiers. Small random noise was introduced to each ITE and ITEs were standardized to have a mean of 0 and a standard deviation of 0.1. Clustering was performed using the Ward.D2 method with the Euclidean distance, evaluating 2–10 clusters. To determine the optimal number of clusters, we employed Beale's index with a significance level set at 0.01. The z-scores of covariates across clusters were computed, and variable importance was determined based on the absolute difference in covariate z-scores. Variables exceeding a predefined threshold of 0.25 were considered potential effect modifiers. Subsequently, we compared the characteristics of the clustered groups after calculating the conditional average treatment effects (CATEs) for each group.

Missing values in all variables were imputed using random forest imputation [34,35]. All analyses were conducted using R (version 4.4.2).

## Sensitivity analyses

To assess the robustness of the variable selection process, we additionally conducted two types of sensitivity analyses. First, we used alternative thresholds of 0.20 and 0.30 for defining potential effect modifiers. Second, we calculated E-values to evaluate the potential impact of unmeasured confounding, which quantify the minimum strength of association that an unmeasured confounder would need to have with both the exposure and the outcome to fully explain the observed associations [36].

## Ethics and data privacy

This study was approved by the Institute of Transdisciplinary Sciences Ethics Committee, Kanazawa University (Approval number: R6-003). Routinely collected data were linked and anonymized by the program provider (PREVENT, Inc.) in accordance with its data privacy policy and relevant regulations. For participants enrolled in the mHealth-based disease management program, agreement to the in-app privacy policy at registration constituted consent for the use of anonymized program data in research. For individuals who did not enroll in the program, only data permitted for secondary use under the commissioned disease prevention service agreement with the client health insurance association were included in the research database, and only anonymized health checkup and claims data were made available to the researchers. All data presented were anonymized, and there are no images or other information in this manuscript that could directly identify individual participants.

## Results

Table 2 presents the baseline characteristics of study participants by treatment status. Participants in the treatment group were more likely to be male, diagnosed with diabetes or prescribed related medication in the past 12 months, have experienced weight changes since age 20, intend to improve lifestyle habits, be older, have a higher body mass index, and exhibit poorer health status compared to those in the control group.

For the ATE of the mHealth-based disease management program on uncontrolled hypertension, participation in the program was associated with a 5.2% (95% confidence interval [CI]: 4.4% to 6.0%) lower risk of uncontrolled hypertension, corresponding to a risk ratio of 0.81 (95% CI: 0.78 to 0.84) compared with conventional treatment alone. This absolute

**Table 2. Baseline characteristics of the study participants.**

| | | Control | Treated | P value |
|---|---|---|---|---|
| **N** | | **2,965** | **127** | |
| Sex (%) | Male | 1,767 (60) | 99 (78) | < 0.01 |
| | Female | 1,198 (40) | 28 (22) | |
| Diagnosed with hypertension or prescribed medication in the past 12 months (%) | Yes | 2,054 (69) | 83 (65) | 0.40 |
| | No | 911 (31) | 44 (35) | |
| Diagnosed with diabetes or prescribed medication in the past 12 months (%) | Yes | 1,147 (39) | 83 (65) | < 0.01 |
| | No | 1,818 (61) | 44 (35) | |
| Diagnosed with dyslipidemia or prescribed medication in the past 12 months (%) | Yes | 1,888 (64) | 86 (68) | 0.40 |
| | No | 1,077 (36) | 41 (32) | |
| Smoking status (%) | Yes | 625 (21) | 24 (19) | 0.55 |
| | No | 2321 (78) | 103 (81) | |
| | Missing | 19 (1) | 0 (0) | |
| Drinking status (%) | Everyday | 705 (24) | 35 (28) | 0.54 |
| | Sometimes | 858 (29) | 35 (28) | |
| | Rare or non-drinkers | 1,298 (44) | 55 (43) | |
| | Missing | 104 (4) | 2 (2) | |
| Weight change since age 20 (%) | Yes | 1,583 (53) | 81 (64) | 0.05 |
| | No | 1,273 (43) | 44 (35) | |
| | Missing | 109 (4) | 2 (2) | |
| Intention to improve lifestyle habits (%) | Yes | 2,384 (80) | 115 (91) | 0.02 |
| | No | 461 (16) | 9 (7) | |
| | Missing | 120 (4) | 3 (2) | |
| Engaging in exercise that causes light sweating for at least 30 minutes per session, at least twice a week, for over a year (%) | Yes | 608 (21) | 35 (28) | 0.09 |
| | No | 2,250 (76) | 90 (71) | |
| | Missing | 107 (4) | 2 (2) | |
| Engaging in walking or equivalent physical activity for at least one hour per day in daily life (%) | Yes | 1,362 (46) | 51 (40) | 0.14 |
| | No | 1,494 (50) | 74 (58) | |
| | Missing | 109 (4) | 2 (2) | |
| Eating speed (%) | Fast | 1,160 (39) | 53 (42) | 0.80 |
| | Normal | 1,527 (52) | 64 (50) | |
| | Slow | 162 (5) | 7 (6) | |
| | Missing | 116 (4) | 3 (2) | |
| Having dinner within two hours before bedtime (%) | Yes | 1,116 (38) | 46 (36) | 0.39 |
| | No | 1,738 (59) | 79 (62) | |
| | Missing | 111 (4) | 2 (2) | |
| Skipping breakfast three or more times per week (%) | Yes | 598 (20) | 22 (17) | 0.25 |
| | No | 2,248 (76) | 103 (81) | |
| | Missing | 119 (4) | 2 (2) | |
| Getting sufficient rest through sleep (%) | Yes | 1,718 (58) | 78 (61) | 0.37 |
| | No | 1,133 (38) | 47 (37) | |
| | Missing | 114 (4) | 2 (2) | |
| Age [mean (SD)] | | 50.7 (8.2) | 52.8 (6.9) | < 0.01 |
| Body mass index [mean (SD)] | | 25.7 (4.9) | 26.7 (3.7) | 0.03 |
| Triglycerides [mean (SD)] | | 127.3 (96.1) | 153.2 (85.0) | < 0.01 |
| HDL cholesterol [mean (SD)] | | 60.8 (17.4) | 54.7 (13.3) | < 0.01 |
| LDL cholesterol [mean (SD)] | | 122.0 (32.7) | 128.4 (30.5) | 0.03 |

*(Continued)*

**Table 2.** (Continued)

|  |  | Control | Treated | P value |
|---|---|---|---|---|
| **N** |  | **2,965** | **127** |  |
| Diastolic blood pressure [mean (SD)] |  | 81.2 (12.5) | 83.3 (12.8) | 0.06 |
| Systolic blood pressure [mean (SD)] |  | 128.9 (17.8) | 130.5 (17.4) | 0.33 |
| Glutamate Oxaloacetate Transaminase [mean (SD)] |  | 26.7 (14.6) | 28.5 (15.3) | 0.18 |
| Glutamate Pyruvate Transaminase [mean (SD)] |  | 32.7 (26.1) | 39.8 (28.0) | < 0.01 |
| Gamma-GTP [mean (SD)] |  | 49.8 (72.5) | 64.3 (71.0) | 0.03 |
| Hemoglobin A1c [mean (SD)] |  | 6.0 (1.1) | 6.7 (1.2) | < 0.01 |
| Charlson Comorbidity Index [mean (SD)] |  | 1.5 (1.7) | 2.2 (1.6) | < 0.01 |
| Number of days since the last health check-up [mean (SD)] |  | -203.2 (108.0) | -198.1 (111.0) | 0.61 |

SD, standard deviation; HDL, high-density lipoprotein; LDL, low-density lipoprotein. Percentages may not sum to exactly 100% due to rounding.

risk reduction corresponds to an NNT of 19.2 (95% CI: 16.7 to 22.7) over the study's follow-up period. The calculated E-value for this association was 1.77. In addition, participation in the program was associated with a mean change in systolic blood pressure of −1.22 mmHg (95% CI: −1.58 to −0.87) and in diastolic blood pressure of −3.67 mmHg (95% CI: −4.14 to −3.22). Fig 2 illustrates the distribution of ITEs for the mHealth-based disease management program on uncontrolled hypertension (median = −0.06; interquartile range [IQR] = 0.25). In our study, the optimal number of clusters was determined to be two. Fig 3 presents the variable importance estimated using the clustering approach which identified 10 potential effect modifiers: intention to improve lifestyle habits, smoking status, diastolic blood pressure, drinking status (merely or never), Glutamate Oxaloacetate Transaminase, engagement in walking, Gamma-GTP, engagement in light sweating exercise, age, and drinking status (sometimes). The optimal number of clusters was determined to be two groups.

Table 3 compares the baseline characteristics of the high-benefit and low-benefit clusters based on covariates exceeding the predefined variable importance threshold of 0.25. The mHealth-based disease management program was more effective for individuals who intended to improve lifestyle habits, were nonsmokers, had higher diastolic blood pressure, were occasional drinkers or non-drinkers, had lower Glutamate Oxaloacetate Transaminase, engaged in walking but not in light sweating exercise, had lower Gamma-GTP levels, and were younger. In the high-benefit cluster, the mean CATE was −0.17 (95% CI: −0.19 to −0.15), while in the low-benefit cluster, the mean CATE was 0.10 (95% CI: 0.08 to 0.12). The corresponding risk ratios were 0.41 (95% CI: 0.34 to 0.48) for the high-benefit cluster and 1.35 (95% CI: 1.28 to 1.42) for the low-benefit cluster. The absolute risk reduction for the high-benefit cluster corresponds to an NNT of 5.9 (95% CI, 5.3 to 6.7) over the study's follow-up period. The E-values for these estimates were 4.31 and 2.04, respectively.

S1 and S2 Tables present the results of our secondary outcomes, comparing the baseline characteristics of the high-benefit and low-benefit clusters based on the covariates selected in the primary outcome analysis. For the secondary outcome analysis, the mean CATE for systolic blood pressure was −8.34 mmHg (95% CI: −8.85 to −7.83) in the high-benefit cluster and 5.38 mmHg (95% CI: 5.09 to 5.68) in the low-benefit cluster. For diastolic blood pressure, the mean CATE was −11.12 mmHg (95% CI: −11.52 to −10.72) in the high-benefit cluster and 5.52 mmHg (95% CI: 5.14 to 5.90) in the low-benefit cluster. S3 and S4 Tables show the results of our sensitivity analyses in which we changed the threshold of 0.20 and 0.30 for defining potential effect modifiers.

## Discussion

This study found that participation in a mHealth-based disease management program was associated with a lower risk of uncontrolled hypertension over approximately one year of follow-up. Additionally, substantial heterogeneity in treatment

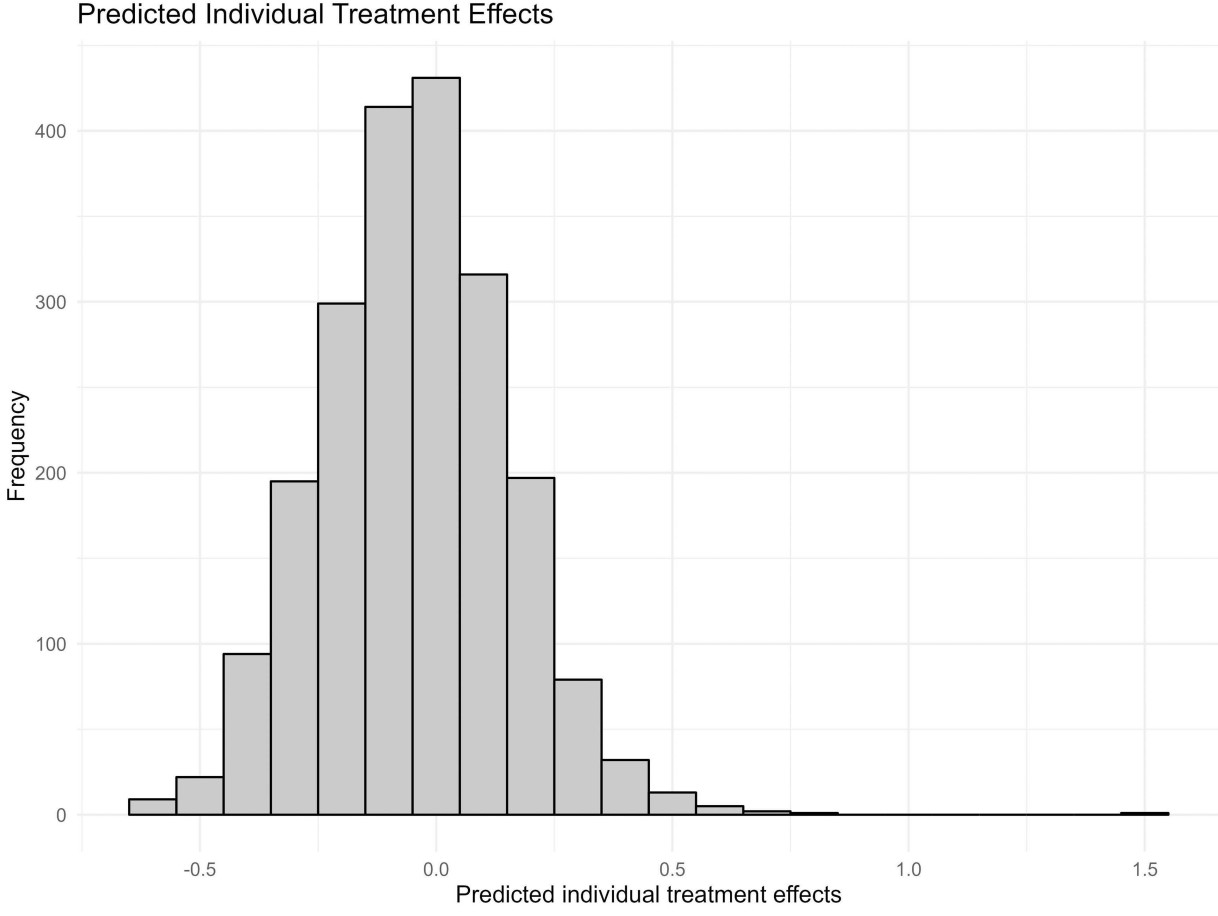

**Fig 2. Histogram of the predicted individual treatment effect estimated via the G-formula.**

response was observed, highlighting the importance of individualized approaches. The extended follow-up period provided valuable insight into the sustained effects and variability across subgroups. Specifically, individuals who benefited the most were those with an intention to improve lifestyle habits, nonsmokers, those with higher diastolic blood pressure, individuals who engaged in walking but not in strenuous exercise, those with lower liver enzyme levels (Glutamate Oxaloacetate Transaminase, Gamma-GTP), younger participants, and occasional drinkers. These findings highlight the potential of digital health interventions for targeted subgroups and emphasize the need for tailored strategies to maximize effectiveness.

Previous systematic reviews and meta-analyses have shown that digital health interventions, including SMS text messages, smartphone applications, and web-based platforms, reduce systolic and diastolic blood pressure [11–20]. However, the reported effects vary, with systolic reductions ranging from approximately 3–7 mmHg and diastolic reductions from 1 to 3 mmHg, depending on study design, target population, and intervention characteristics [12,13,18,20]. While minimal clinically important differences for blood pressure reduction in digital interventions remain unclear, our study employed a pragmatic trial design using uncontrolled hypertension—defined as systolic ≥140 mmHg or diastolic ≥90 mmHg—as a clinically meaningful endpoint. Based on this definition, participation in the mHealth-based disease management program resulted in a 5.2% absolute reduction in the prevalence of uncontrolled hypertension compared with conventional treatment. This approach enhances the real-world applicability of our findings and aligns the evaluation of digital interventions

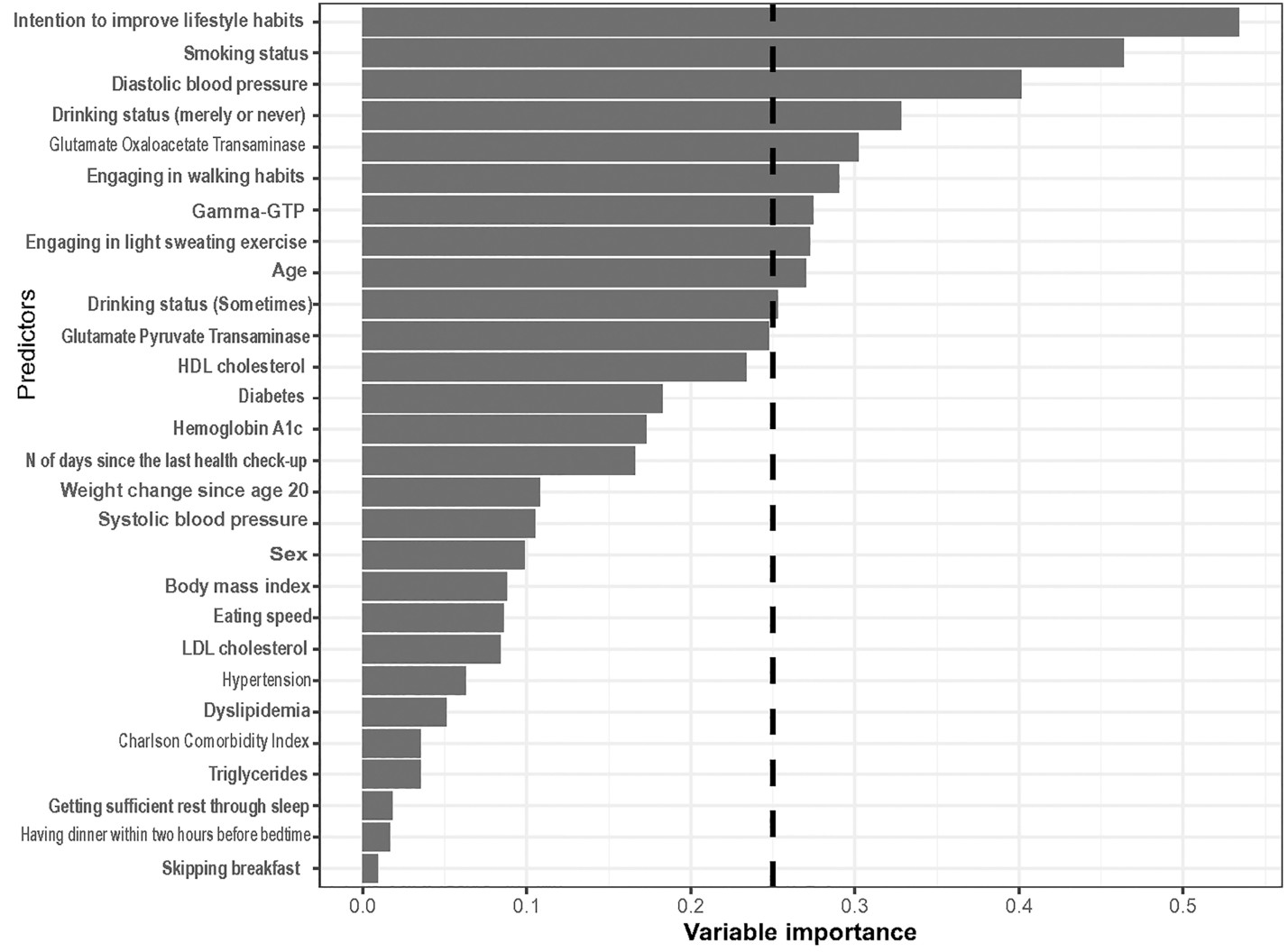

**Fig 3. Variable importance estimated via the cluster modelling.** Only the top covariates with an importance score of 0.25 or higher were considered potential effect modifiers of the treatment effect on the outcome.

with established clinical thresholds. Consistent with this threshold-based endpoint, secondary analyses of continuous blood pressure values showed mean changes in systolic and diastolic blood pressure (−1.22 and −3.67 mmHg, respectively). These mean reductions are compatible with the observed decrease in uncontrolled hypertension and should be interpreted alongside the substantial heterogeneity in individual treatment responses described below.

Treatment effect heterogeneity has been reported in previous studies of digital health interventions, with some suggesting that younger individuals and those with poorly controlled baseline blood pressure benefit more from such programs [11,15,21]. Our findings partially align with this, as individuals with higher diastolic blood pressure and younger age were more likely to experience benefits. However, in our study, younger age played a modest role compared with modifiable factors such as intention to improve lifestyle habits, lifestyle behaviors (e.g., walking habits, smoking status), and metabolic characteristics (Glutamate Oxaloacetate Transaminase, Gamma-GTP). These behavioral and physiological profiles may reflect a population with moderate health awareness and partial engagement in preventive behaviors—individuals

**Table 3. Cluster effects and the differences in baseline characteristics between individuals with high vs. low individual treatment effect.**

| Clusters | CATE (95% CIs) | Proportion Without Intention to Improve Lifestyle Habits | Proportion of Current Smokers | Diastolic Blood Pressure (mean) | Proportion of Non-Drinkers or Rare Drinkers | Glutamate Oxaloacetate Transaminase (mean) | Proportion of Not Engaging in Walking | Gamma-GTP (mean) | Proportion of Not Engaging in Exercise that Causes Light Sweating (≥30 min) | Age (mean) | Proportion of Occasional Drinkers |
|---|---|---|---|---|---|---|---|---|---|---|---|
| High-benefit | −0.17 (−0.19 to −0.15) | 0.08 | 0.71 | 82.91 | 0.39 | 24.70 | 0.45 | 40.64 | 0.85 | 49.15 | 0.34 |
| Low-benefit | 0.10 (0.08 to 0.12) | 0.28 | 0.90 | 77.98 | 0.56 | 28.98 | 0.60 | 62.36 | 0.75 | 51.35 | 0.22 |

CATE, conditional average treatment effect; CI, confidence interval.

who are not entirely inactive but have room for improvement. For example, nonsmokers and occasional drinkers may already exhibit baseline health consciousness, making them more responsive to healthcare guidance. Similarly, participants who walk regularly but do not engage in moderate-to-vigorous exercise may represent an intermediate activity level, where incremental behavior changes lead to measurable benefits [37]. Taken together, these patterns suggest that the high-benefit cluster represents individuals with an intermediate activity baseline who are particularly receptive to low-barrier strategies, such as app-based monitoring and step-goal prompts, rather than prescriptions for vigorous exercise. The elevated diastolic blood pressure observed among high-benefit individuals could also indicate greater physiological potential for improvement. Collectively, these findings suggest that digital health interventions may be particularly effective for individuals with partial readiness and modifiable risk factors [38], offering a key opportunity for targeted and efficient support. Nonetheless, the same behavioral readiness and engagement capacity that characterize the high-benefit cluster may also predispose these individuals to greater adherence to the mHealth program (e.g., more consistent app use and responsiveness to coaching). As our heterogeneity analysis relied solely on baseline covariates and did not explicitly model time-varying adherence or behavior change, part of the observed between-cluster differences in treatment effects may therefore reflect self-selection into higher-engagement patterns, in addition to genuine differences in causal responsiveness.

In contrast, participants in the low-benefit cluster exhibited a higher probability of uncontrolled hypertension (CATE = 0.10). This positive CATE should not be interpreted as evidence that the program was harmful, but rather as indicating a subgroup with limited responsiveness to low-intensity, app-based coaching. Individuals in this cluster may have more complex comorbidity profiles, lower adherence to lifestyle recommendations or antihypertensive treatment, or psychosocial and socioeconomic challenges that were not fully captured by baseline covariates. For such patients, more intensive, multimodal, or face-to-face support beyond the present mHealth-based disease management program may be required to achieve meaningful improvements in blood pressure control. This observed heterogeneity may reflect differences in health engagement capacity or physiological responsiveness to behavioral interventions, highlighting the need for adaptive intervention strategies tailored to individual readiness and biological context. Consistent with the recently proposed high-benefit approach in precision medicine, which emphasizes targeting individuals who are most likely to benefit rather than those who are merely at highest risk [39], our findings illustrate how data-driven estimation of heterogeneous treatment effects can help prioritize mHealth-based support for subpopulations with greater expected gains while also identifying groups with limited benefit who may require alternative intervention strategies.

Given these insights, digital health interventions should be designed with greater flexibility to accommodate varying levels of engagement and behavioral readiness. Prior research suggests that motivation plays a critical role in the

effectiveness of such programs, with higher engagement consistently linked to greater improvements in physical activity, dietary choices, and weight loss [40,41]. Additionally, tailored interventions incorporating goal setting, personalized feedback, and self-monitoring tools have demonstrated superior outcomes [42]. Our findings align with this evidence, highlighting the importance of behavioral readiness—particularly the intention to improve lifestyle habits—in determining intervention effectiveness. Participants with strong motivation to change their health behaviors experienced greater benefits, suggesting that engagement-driven strategies may enhance intervention success [43]. For highly motivated individuals, interactive coaching and personalized feedback may further improve adherence and maximize outcomes [11]. However, for those with lower readiness for behavior change, alternative approaches such as passive tracking via mobile apps or wearable devices may serve as an initial engagement tool, gradually fostering sustained behavioral improvements [44].

A key strength of this study lies in its use of a three-step, data-adaptive framework that integrates outcome regression via the G-formula with SuperLearner-based model specification to estimate treatment heterogeneity. By combining the G-formula with a SuperLearner ensemble model, we can assess individual-level treatment effects while minimizing overfitting and preserving interpretability. This approach extends beyond traditional ATE estimates, providing a more nuanced understanding of how different subgroups respond to mHealth-based interventions [45]. Moreover, the consistent direction of effects between the primary binary endpoint and the secondary continuous outcomes supports the robustness of the estimated heterogeneous treatment effects. Such granularity may inform the development of more effective, personalized strategies for hypertension management. From a methodological perspective, this framework illustrates how ensemble machine learning can enhance individualized decision-making in chronic disease management. Integrating causal inference with data-adaptive modeling offers a transparent and scalable pathway for optimizing personalized digital health interventions. Future studies should evaluate similar approaches in more robust baseline balancing, stratification analyses, and stricter clinical endpoints such as major cardiovascular adverse events.

Additionally, this study addresses a critical gap in the literature regarding the long-term impact of digital health interventions. While many previous trials have been limited to follow-up periods of 3–12 months [12,18–20], our analysis used a one-year follow-up period within a real-world setting. This extended observation window allowed for the evaluation of both sustained effects and treatment heterogeneity over time, offering valuable insights into the durability and subgroup variability of intervention outcomes. Such long-term evidence is essential for guiding future implementation strategies in chronic disease management.

This study has several limitations. First, although the target trial emulation framework enabled long-term evaluation, follow-up duration differed across participants, which may have introduced inconsistency in exposure and outcome measurements. Second, while the mHealth-based disease management program collected behavioral and clinical data throughout the intervention, our analysis relied solely on baseline characteristics to estimate treatment effects. As a result, changes in health behaviors or levels of program engagement over time could not be incorporated, and adherence-related mechanisms were not examined. Third, while our findings suggest that certain behavioral and metabolic characteristics are associated with intervention effectiveness, the generalizability of these results may be limited to populations with similar demographic and occupational backgrounds. Fourth, individuals with chronic complications may exhibit lower adherence to both therapeutic regimens and mHealth-based interventions, potentially leading to an underestimation of the observed intervention effects. Moreover, observed heterogeneity may partially reflect self-selection mechanisms and differential capacity for engagement or adherence, rather than true causal effect modification. Fifth, baseline characteristics differed substantially between the treated and untreated groups, raising concerns about the identifiability of causal effects. Although the G-computation approach adjusts for measured confounders through model-based standardization, its validity depends on the positivity assumption. To evaluate this, we examined the empirical distribution of estimated treatment probabilities and observed a limited but clearly non-zero region of overlap between groups (S1 Fig). While empirical positivity was restricted but not violated, this limited overlap constrains individual-level causal interpretation, particularly for heterogeneous treatment effect estimation. Finally, as with all observational studies, the potential for unmeasured confounding

remains, despite the application of robust causal inference techniques. In particular, information on medication use, adherence patterns, and socioeconomic or behavioral factors, such as digital literacy and access to technology, were not available and could not be adjusted for. Furthermore, several baseline differences between groups may have partially influenced the observed associations. We addressed this potential confounding using G-computation to obtain marginal estimates adjusted for baseline covariates, and our calculated E-values indicated moderate robustness to unmeasured confounding [46]. The ITE estimates should therefore be used to support clinical decision-making and prioritization, not to inform eligibility or exclusion rules for digital health interventions. Accordingly, our findings are best interpreted as illustrating how target trial emulation can be applied to real-world data to evaluate mHealth-based disease management programs, while providing supportive evidence that such programs may be associated with better blood pressure control in routine care.

## Conclusion

Using a target trial emulation framework with approximately one year of follow-up data, this study found that participation in an mHealth-based disease management program was associated with a 5.2% reduction in the prevalence of uncontrolled hypertension compared with conventional treatment. Importantly, substantial variability in treatment response was observed across individuals. These findings highlight the value of extended follow-up in capturing sustained effects and suggest that treatment response may be more closely linked to modifiable behavioral characteristics than to fixed demographic traits. Digital health interventions may offer particular benefits for individuals with adaptable risk profiles. Future research should explore how comprehensive baseline assessments can guide the personalization of digital health programs to better align with individual readiness and risk, thereby enhancing their long-term effectiveness and broader applicability.

## Supporting information

**S1 Fig. The empirical distribution of estimated treatment probabilities.**
(DOCX)

**S1 Table. Cluster effects and the differences in baseline characteristics between individuals with high vs. low individual treatment effect (Secondary outcome=difference in diastolic blood pressure between baseline and follow-up).**
(DOCX)

**S2 Table. Cluster effects and the differences in baseline characteristics between individuals with high vs. low individual treatment effect (Secondary outcome=difference in systolic blood pressure between baseline and follow-up).**
(DOCX)

**S3 Table. Sensitivity analysis with the variable importance threshold set to 0.20.**
(DOCX)

**S4 Table. Sensitivity analysis with the variable importance threshold set to 0.30.**
(DOCX)

## Acknowledgments

We would like to express our sincere gratitude to the participants who contributed to this study. We are also deeply grateful to Mr. Kojiro Yamamoto, formerly with PREVENT Inc., for his dedicated support and contributions during the implementation of this project. Professional English language editing was performed by Enago to ensure clarity and accuracy.

Transparency statement: The authors used ChatGPT only as a writing aid to help refine the English language and improve the clarity of early manuscript drafts. No new scientific content, study design, data analysis, or conclusions were generated by the tool. All text was reviewed, edited, and approved by the authors, who take full responsibility for the final manuscript.

## Author contributions

**Conceptualization:** Takahiro Miki, Takuya Toda, Yuta Hagiwara.

**Data curation:** Takahiro Miki, Takuya Toda, Yuta Hagiwara.

**Formal analysis:** Takaaki Ikeda.

**Funding acquisition:** Yuta Hagiwara.

**Methodology:** Masashi Kanai, Takuya Toda.

**Project administration:** Takahiro Miki.

**Resources:** Yuta Hagiwara.

**Supervision:** Takahiro Miki, Takuya Toda, Yuta Hagiwara, Takaaki Ikeda.

**Validation:** Takuya Toda.

**Writing – original draft:** Masashi Kanai, Takaaki Ikeda.

**Writing – review & editing:** Masashi Kanai, Takahiro Miki, Takaaki Ikeda.

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
