## [Decision Letter · Decision Letter 0]

3 Nov 2025

Response to Reviewers
Revised Manuscript with Track Changes
Manuscript
**Journal Requirements:**

1. Please send a completed ‘Competing Interests’ statement, including any COIs declared by your co-authors. Please declare all competing interests beginning with the statement “I have read the journal's policy and the authors of this manuscript have the following competing interests:”

For more information, please go to our submission guidelines:

https://journals.plos.org/digitalhealth/s/submission-guidelines#loc-competing-interests

2. In this instance it seems there may be acceptable restrictions in place that prevent the public sharing of your minimal data. However, in line with our goal of ensuring long-term data availability to all interested researchers, PLOS’ Data Policy states that authors cannot be the sole named individuals responsible for ensuring data access (http://journals.plos.org/digitalhealth/s/data-availability#loc-acceptable-data-sharing-methods).

For further assistance, you may go to: http://journals.plos.org/digitalhealth/s/data-availability

3. Please provide separate main figure files in .tif or .eps format only and ensure that all files are under our size limit of 10MB.

For more information about how to convert your figure files please see our guidelines: https://journals.plos.org/digitalhealth/s/figures

**Additional Editor Comments (if provided):**
**Reviewers' Comments:**

**Comments to the Author**

1. Does this manuscript meet PLOS Digital Health’s publication criteria?

Reviewer #1: Yes

Reviewer #2: Partly

Reviewer #3: Partly

Reviewer #4: Yes

Reviewer #5: No

2. Has the statistical analysis been performed appropriately and rigorously?

Reviewer #1: Yes

Reviewer #2: Yes

Reviewer #3: No

Reviewer #4: No

Reviewer #5: No

3. Have the authors made all data underlying the findings in their manuscript fully available (please refer to the Data Availability Statement at the start of the manuscript PDF file)?

Reviewer #1: Yes

Reviewer #2: Yes

Reviewer #3: Yes

Reviewer #4: Yes

Reviewer #5: No

4. Is the manuscript presented in an intelligible fashion and written in standard English?

Reviewer #1: Yes

Reviewer #2: Yes

Reviewer #3: No

Reviewer #4: Yes

Reviewer #5: Yes

Reviewer #1: 1. The manuscript numbers are rather specific to the submission process for the journal, which is fine, but the Short Title is duplicated. Ensure that it appears only once or as necessary by the final layout.

2.The character of the "low-benefit cluster" (CATE of 0.10) is presented without a deep interpretive understanding. That the intervention increased this group's risk of uncontrolled hypertension by 10% needs to be talked about with care, particularly in the context of traditional treatment alone.

3. The finding that the high-benefit group walked but didn't engage in light sweating exercise may be deceptive. This is a distinction which must be clarified in the Discussion—is the group receptive to low-barrier strategies, or is program design beneficial for those with an intermediate activity baseline?.

4. The accepted limitation of relying only on baseline characteristics to assess ITE, not counting changes in health behaviors or program participation during the six-month intervention, is a significant limitation for discerning heterogeneity fully.

5. In the Discussion, devote a paragraph to interpreting the CATE of 0.10 for the low-benefit cluster. An affirmative CATE signifies increased risk of uncontrolled hypertension. Provide a conservative, evidence-based description. Is this possibly a population with such suboptimal adherence/comorbidities that low intensities of remote coaching are insufficient or disruptive to more pressing intense care (usual treatment)?. Is it possible that this population has problems (e.g., psychological, socioeconomic) not accounted for by baseline covariates and thus have poor outcomes regardless of the intervention?

6. It is suggested that the authors to include about AI/ML in decision support or healthcare optimization because the approach largely depends on machine learning (SuperLearner, clustering) to enhance decision-making (personalized treatment). The approach in the manuscript utilizes ensemble machine learning in estimating individual treatment effects, which is essentially an improvement and optimization problem of decision-making. The suggested paper from the reference: "Blockchain-Enabled Healthcare Optimization: Enhancing Security and Decision-Making Using the Mother Optimization Algorithm" this paper discusses utilizing advanced algorithms to improve decision-making and enable safe, transparent, and efficient medical data handling. This is in line most particularly with the mission of the manuscript to utilize ML for assisting with enhanced treatment choice.

7. The authors must react to the following main points: Low-Benefit Cluster Interpretation: Describe the interpretation of the CATE of 0.10 under Discussion. Provide a cautious, soundly argued reason why the intervention negatively affected this subgroup (e.g., condition complexity, non-adherence, low level of intervention). Clearness on Exercise: Clarify the definition of high-benefit cluster walking but not in light sweating exercise to clearly say the clinical/behavioral implication: more likely to be a change ready group and would benefit from low-to-moderate intensity digital nudges.

Reviewer #2: Kanai and colleagues explored the heterogeneous effects of a mobile health-based disease management program on uncontrolled hypertension. The research topic is interesting, but the findings are undermined by the study design.

Major comments:

1. Major concerns come from the huge disparities in baseline characteristics of the study participants between Control group and Treated group (Table 2). Key confounders such as sex, diabetes, age, triglycerides and so on are imbalanced between the two groups. The Treated group mostly has more severe disease conditions than the Control group, making the target trial emulation not reliable.

2. In Table 1, the Inclusion criteria are ambiguous. If the authors aim to explore the impact of mHealth on hypertension, they should focus on those with hypertension undergoing drug therapies. It is unclear why the authors equally included the patients with diabetes and dyslipidaemia. The authors also excluded patients with “severe” cardiac diseases such as arrhythmias, though arrhythmias such as atrial fibrillation are quite common in those patients with hypertension. It is not quite clear what is “severe”.

3. The study Outcome of uncontrolled hypertension may be better replaced by actual blood pressure values or the difference in blood pressure between the baseline and follow-up.

4. Blood pressure was assessed annually during the health checkup. Is it only one measurement? How was it performed? How is the reliability of this measurement if it was only measured once?

Minor comments:

1. In the Treatment arm, did patients follow all three programs?

2. The authors said that they followed the STROBE guideline, so a checklist should be provided.

3. In “Inclusion and exclusion criteria” section, the Guithub link for the exclusionary diagnoses is not accessible. The authors may check it.

Reviewer #3: - The paper claims that participation in a mobile health–based disease management program was associated with a lower prevalence of uncontrolled hypertension after one year, and that this effect varied across individuals. It applies target trial emulation with causal machine learning to estimate both average and individual treatment effects.

These claims are important and timely for digital health, as they promote rigorous causal methods and personalized intervention evaluation in real-world data. However, the causal inference remains moderate given the observational design and limited sensitivity analyses.

- The claims are only partially novel. The same dataset and blood pressure thresholds were analyzed in (J Med Internet Res 2023;25:e43809) doi: 10.2196/43809, which already demonstrated improved BP control with the mHealth app. The current manuscript mainly reframes the same study under a target-trial-emulation approach and adds heterogeneity analysis using causal ML. Thus, it contributes methodologically rather than substantively and could be considered complementary research rather than fully original work.

- Line 118: website is not working

- In Methods, Table 1 (“Treatment assignment”), replace “Individuals were randomly allocated following the eligible assessment” with “Treatment assignment was emulated analytically within the target trial framework.”

- Covariate-balance diagnostics: To align with current Target Trial Emulation guidance, please demonstrate that the emulated treatment and control groups were comparable at baseline. Add a Supplementary Figure or table showing standardized mean differences (SMDs) for all baseline covariates before and after adjustment or weighting.

- Sensitivity analyses for unmeasured confounding: Please add a short subsection titled “Sensitivity analyses” in the Methods and present corresponding results. Mention in the Discussion how sensitive or robust the main findings are to moderate unmeasured confounding. This addition is required to support the validity of the causal interpretation.

- The main outcome is currently binary (“uncontrolled hypertension” yes/no). To enhance clinical interpretability, please also report continuous BP outcomes (ΔSBP and ΔDBP in mmHg, mean ± SD and 95 % CI).

- Missing covariates: number of medications, adherence patterns, socioeconomic and behavioral factors (include digital literacy/technology access)

- Table 3: The reported CATE values (−0.17 vs 0.10) lack measures of precision (e.g., 95 % CIs or standard errors). The predefined variable importance threshold of 0.25 appears arbitrary. Please justify the choice or, alternatively, present sensitivity analyses using different thresholds (e.g., 0.20, 0.30) to demonstrate robustness.

- The Discussion (line 252-272) nicely interprets the observed heterogeneity in treatment effects in terms of behavioral readiness and motivation. While the discussion highlights behavioral readiness and engagement capacity, it would be useful to also acknowledge that these same factors might contribute to self-selection into higher adherence groups, potentially inflating apparent heterogeneity.

Grammar and Readability:

Hyphenation: six-month, one-year, follow-up, data-adaptive, SuperLearner-based.

Terms: uncontrolled hypertension (avoid “insufficient hypertension control”).

Replace causal verbs in observational context: “was associated with” instead of “reduced,” “improved,” “caused.”

“modelling” → “modeling” (or choose one variant and use it throughout).

Line 35: was associated with a 5.2%

Covariates (Line 145): “engaging in moderate exercise (causing light sweating) …”;

“rare or non-drinkers” (instead of “merely or never”).

Line 207: “the intervention reduced the risk…” → “was associated with a 5.2% lower risk/prevalence…”, same in Discussion first paragraph

Reviewer #4: The topic is relevant, the statistical design is robust, and the use of the G-formula method combined with the SuperLearner algorithm is innovative. The study benefits from being conducted within Japan’s well-structured health system, which provides equitable access to healthcare. However, several methodological aspects require clarification: there are statistically significant baseline differences between groups, and there is a bias related to lower adherence among patients with chronic or cardiac diseases. An individual-level analysis is needed to determine the weight and influence of each baseline variable on the final model. The writing is clear and well organized, though causal expressions such as “reduced” should be replaced with inferential phrasing like “was associated with.” In the abstract and results, it should be emphasized that baseline differences, while small in magnitude, were statistically significant and may have influenced the findings, as the two groups represent populations with different baseline profiles. The authors should also clarify that the assignment was voluntary and observational, not randomized, since the selection criteria and baseline imbalances (age, BMI, triglycerides, HDL, HbA1c, Charlson index) prevent the study from being considered experimental; these imbalances could bias effect estimation. A sensitivity or variable importance analysis (for example, variable importance plots, multivariable regression, or partial dependence analysis) is recommended to quantify the individual influence of each factor on the final model, along with stratified analyses by sex and the presence of cardiovascular disease. It should be acknowledged that patients with chronic complications often show lower therapeutic and digital adherence, which could attenuate the observed effects. The main clinical finding—an absolute reduction of 5.2%, or about 52 fewer uncontrolled cases per 1,000 treated patients (NNT ≈ 19)—should be expressed in clinical terms, while noting that adherence and selection biases may have contributed to the observed associations, as patients who comply with follow-up are also more likely to adhere to treatment, a correlated but non-causal phenomenon. The study aligns with ESC/ESH 2023 and ACC/AHA 2024 hypertension guidelines supporting mHealth use and would benefit from future research with better baseline balance, stratified analyses, and harder clinical endpoints (MACE). Minor revisions include defining all abbreviations upon first mention, standardizing numerical and p-value formatting, and moving the ChatGPT mention to a Transparency Statement. Overall, the study is relevant and well structured but requires a Major Revision to: (1) acknowledge and quantify statistically significant baseline differences, (2) perform an individual-level analysis to assess the weight of each variable on the final model, and (3) strengthen sensitivity analyses to ensure causal validity. Once these revisions are incorporated, the manuscript will provide robust and clinically meaningful evidence of the personalized benefits of digital interventions for hypertension control.

Reviewer #5: The study is promising in concept and explores a relevant topic - long-term effectiveness and personalized responses to mHealth hypertension interventions. However, it currently lacks the methodological rigor and transparency required for publication. Some areas of key improvement are to be noted:

1. please provide a clear description of the randomization process and timing. The manuscript claims random allocation but lacks specific details on the randomization process (eg. method of sequence generation, allocation concealment), making it unclear if true randomization occurred.

2. The study design does not fully address the risk of selection bias, especially given the retrospective use of a commercial database and imbalances in baseline characteristics between groups suggest this may be a concern

3. no mention of blinding for participants, assessors, or data analysts, which could introduce bias in outcomes assessment and data interpretation.

4.The statistical methods used are not fully explained and the use of only summary measures (eg. means, medians) without individual data points raises concerns regarding the robustness and reliability of the analysis.

5. please provide clear details on ethical approval, consent, or participant privacy consideration

6. please avoid the use of terms like "significant" without providing context in terms of statistical analysis/p-values

7. the sample imbalance (127 vs 2,965) and lack of adjustment through weighting, matching, etc estimators leave open confounding concerns

8. Include sensitivity analyses using continuous BP outcomes and contextualize results against known clinical thresholds. For eg, the 'absolute risk reduction of 5.2%' lacks clinical context. What does this correspond to in mmHg reduction?

**Do you want your identity to be public for this peer review?** For information about this choice, including consent withdrawal, please see our Privacy Policy

Reviewer #1: No

Reviewer #2: No

Reviewer #3: No

Reviewer #4: **Yes:** Adan Pacifuentes Orozco

Reviewer #5: No

**Figure resubmission:**

**Reproducibility:** To enhance the reproducibility of your results, we recommend that authors of applicable studies deposit laboratory protocols in protocols.io, where a protocol can be assigned its own identifier (DOI) such that it can be cited independently in the future. Additionally, PLOS ONE offers an option to publish peer-reviewed clinical study protocols. Read more information on sharing protocols at https://plos.org/protocols?utm_medium=editorial-email&utm_source=authorletters&utm_campaign=protocols

---

## [Decision Letter · Decision Letter 1]

29 Dec 2025

Response to Reviewers
Revised Manuscript with Track Changes
Manuscript
**Journal Requirements:**
**Additional Editor Comments (if provided):**
**Reviewers' Comments:**

**Comments to the Author**

Reviewer #1: All comments have been addressed

Reviewer #3: All comments have been addressed

Reviewer #4: All comments have been addressed

Reviewer #5: All comments have been addressed

publication criteria?

Reviewer #1: Yes

Reviewer #3: Yes

Reviewer #4: Yes

Reviewer #5: Yes

3. Has the statistical analysis been performed appropriately and rigorously?

Reviewer #1: Yes

Reviewer #3: Yes

Reviewer #4: Yes

Reviewer #5: I don't know

4. Have the authors made all data underlying the findings in their manuscript fully available (please refer to the Data Availability Statement at the start of the manuscript PDF file)?

Reviewer #1: Yes

Reviewer #3: Yes

Reviewer #4: No

Reviewer #5: Yes

5. Is the manuscript presented in an intelligible fashion and written in standard English?

Reviewer #1: Yes

Reviewer #3: Yes

Reviewer #4: Yes

Reviewer #5: Yes

Reviewer #1: The authors have revised all the recommendations and suggestions that was suggested earlier.

Reviewer #3: The reviewer has addressed the comments thoroughly and in a clear and specific manner. Overall, after revision, this manuscript is suitable for publication in the journal. I have no further comments.

Reviewer #4: This study estimates the long-term effectiveness of a chronic disease management program based on mHealth for patients with uncontrolled hypertension, a significant pathology with a high burden of morbidity and mortality. A target trial emulation design was chosen, employing causal inference methods and machine learning as an appropriate approach. The topic is highly relevant in the fields of medicine and digital medicine, as it addresses the need for even stronger evidence regarding the sustainability of mHealth interventions in real-world populations and their monitoring. In general, the study is well-structured, methodologically ambitious, and effectively utilizes complex and preferred analytical tools, which strengthens its results, particularly from a methodological standpoint.

The study presents significant structural limitations that must be considered when interpreting the results. The main one is the overall imbalance between the treatment groups, as well as the lack of control over fundamental variables that have an impact, since these in themselves represent well-studied risk factors such as sex, metabolic profile, and comorbidity burden. This generates a significant risk of bias when interpreting the results. Although the authors adequately correct for these biases using the G-formula, showing that positivity is not strongly violated, the limited overlap between groups restricts causal inference at the individual level. Stratified studies should be conducted due to the baseline differences between the groups. We must consider that, although there is no statistical impact, the aforementioned factors are considered individual risk factors in classic, highly standardized studies with rigorous methodology. Added to this is the potential for uncontrolled confounding, especially variables associated with the number and type of antihypertensive medications, medication adherence, socioeconomic status, digital literacy, and actual use of the application. These considerations are considered particularly important for interpreting heterogeneity analyses, since the benefit found may be contaminated by self-selection and greater compliance capacity, rather than causal differences in program response.

We consider the following recommendations for the authors: this study could be further enriched by including longitudinal measures of participation and application use, which would allow for a better separation of the program's causal effect from the adherence-mediated effect—that is, an analysis of adherence. In addition, the aforementioned analyses stratified by sex or pre-existing cardiovascular disease, as well as by other variables, should be included. We consider it important to emphasize that individual treatment effects should be considered as prioritization criteria and aids in clinical decision-making, but never as criteria for excluding patients. From an editorial perspective, the manuscript is enhanced by defining itself as a methodological contribution to improving how digital interventions are evaluated using real-world data, rather than a demonstration of causal efficacy.

The above study is well-designed, innovative, and aligns with current hypertension guidelines that advocate for the complementary use of digital tools to achieve treatment and control. Its results should be interpreted with caution, given the inherent limitations of an observational design, but it can serve as a starting point for developing more personalized digital health strategies. With the necessary corrections and maintaining an interpretive tone, the manuscript is ready for publication.

Reviewer #5: (No Response)

**Do you want your identity to be public for this peer review?** For information about this choice, including consent withdrawal, please see our Privacy Policy

Reviewer #1: No

Reviewer #3: No

Reviewer #4: **Yes:** Adan Pacifuentes Orozco

Reviewer #5: No

**Figure resubmission:**

**Reproducibility:** To enhance the reproducibility of your results, we recommend that authors of applicable studies deposit laboratory protocols in protocols.io, where a protocol can be assigned its own identifier (DOI) such that it can be cited independently in the future. Additionally, PLOS ONE offers an option to publish peer-reviewed clinical study protocols. Read more information on sharing protocols at https://plos.org/protocols?utm_medium=editorial-email&utm_source=authorletters&utm_campaign=protocols

---

## [Decision Letter · Decision Letter 2]

5 Feb 2026

Response to Reviewers
Revised Manuscript with Track Changes
Manuscript
**Journal Requirements:**
**Additional Editor Comments (if provided):**
**Reviewers' Comments:**

**Comments to the Author**

Reviewer #4: All comments have been addressed

publication criteria?

Reviewer #4: Yes

3. Has the statistical analysis been performed appropriately and rigorously?

Reviewer #4: Yes

4. Have the authors made all data underlying the findings in their manuscript fully available (please refer to the Data Availability Statement at the start of the manuscript PDF file)?

Reviewer #4: Yes

5. Is the manuscript presented in an intelligible fashion and written in standard English?

Reviewer #4: Yes

Reviewer #4: I appreciate how carefully and thoughtfully the authors revised the text. The changes make it much clearer and set clear limits on how the data can be interpreted. The longer discussion of baseline imbalance, limited empirical positivity, and possible self-selection and adherence-related mechanisms does a good job of addressing the primary methodological issues that were brought up before.

At this point, I would want to explain in more detail why I suggested changing the title of the manuscript. The work utilizes sophisticated causal inference methodologies, such as target trial emulation, G-computation, and machine learning–based outcome modeling; nonetheless, the foundational data are inherently observational. Consequently, the study is fundamentally constrained in its capacity to directly assess the causal effect of hypertension therapy. Essential aspects of clinical management—including treatment intensification, medication adherence, clinician decision-making, and longitudinal variations in therapy—cannot be entirely observed or regulated within the existing paradigm. As a result, residual confounding and selection mechanisms cannot be completely eliminated, even with strong analytical techniques.

From a methodological standpoint, the results endorse inferences concerning modeled treatment effects and relationships within a causal framework, rather than conclusive causal determinations regarding the efficacy of hypertension care. The primary value of the manuscript is its demonstration of how contemporary causal and data-adaptive methodologies can be utilized to investigate variation in blood pressure outcomes and to guide hypothesis formulation, prioritization, and the design of future studies. However, the existing title might make some readers think that there is a level of causal certainty that an observational study, no matter how rigorous, can't fully support.

For this reason, I think that changing the title is a fair and helpful way to keep the scientific value of the work while making sure that readers understand what the study can really tell them. Putting the words "evaluation," "association," or "methodological assessment" in the title would better show what the study can and can't say, without making it less useful for clinical practice or digital health research.

I advise using titles that focus on evaluation, association, or methodological assessment instead of direct causal consequences of hypertension management to better match the manuscript title with the observational study design and its inferential scope. Some other options are:

Assessment of a mobile health-oriented disease management program for uncontrolled hypertension by target trial emulation.

Blood pressure regulation linked to a mobile health-oriented illness management program: a practical target trial emulation study

Diversity in blood pressure outcomes among participants in a mobile health–based hypertension management program

All of these choices keep the study's scientific contribution and methodological rigor while also making it clear that the publication does not directly quantify the causal effect of managing hypertension, which cannot be fully demonstrated with the current observational design.

**Do you want your identity to be public for this peer review?** For information about this choice, including consent withdrawal, please see our Privacy Policy

Reviewer #4: **Yes:** Adan Pacifuentes Orozco

**Figure resubmission:**

**Reproducibility:** To enhance the reproducibility of your results, we recommend that authors of applicable studies deposit laboratory protocols in protocols.io, where a protocol can be assigned its own identifier (DOI) such that it can be cited independently in the future. Additionally, PLOS ONE offers an option to publish peer-reviewed clinical study protocols. Read more information on sharing protocols at https://plos.org/protocols?utm_medium=editorial-email&utm_source=authorletters&utm_campaign=protocols

---

## [Decision Letter · Decision Letter 3]

10 Feb 2026

Heterogeneous associations of a mobile health-based disease management program on uncontrolled hypertension: a target trial emulation study

PDIG-D-25-00810R3

Dear Mr. Miki,

We are pleased to inform you that your manuscript 'Heterogeneous associations of a mobile health-based disease management program on uncontrolled hypertension: a target trial emulation study' has been provisionally accepted for publication in PLOS Digital Health.

Best regards,

Haleh Ayatollahi

Section Editor

PLOS Digital Health

**Additional Editor Comments (if provided):**

**Reviewer Comments (if any, and for reference):**

Reviewer's Responses to Questions

**Comments to the Author**

Reviewer #4: All comments have been addressed

publication criteria?

Reviewer #4: Yes

3. Has the statistical analysis been performed appropriately and rigorously?

Reviewer #4: Yes

4. Have the authors made all data underlying the findings in their manuscript fully available (please refer to the Data Availability Statement at the start of the manuscript PDF file)?

Reviewer #4: Yes

5. Is the manuscript presented in an intelligible fashion and written in standard English?

PLOS Digital Health does not copyedit accepted manuscripts, so the language in submitted articles must be clear, correct, and unambiguous. Any typographical or grammatical errors should be corrected at revision, so please note any specific errors here.

Reviewer #4: Yes

Reviewer #4: The requested changes have been implemented, and the manuscript is suitable for publication.

**Do you want your identity to be public for this peer review?** For information about this choice, including consent withdrawal, please see our Privacy Policy

Reviewer #4: **Yes:** Adán Pacifuentes Orozco
